# Willingness to pay for improvements in rural sanitation: Evidence from a cross-sectional survey of three rural counties in Kenya

**Diana Mutuku Mulatya**[1¤a]*, **Vincent Were**[2¤b], **Joseph Olewe**[3¤c], **Japheth Mbuvi**[1¤a]

**1** USAID/Kenya Integrated Water, Sanitation and Hygiene Project, Nairobi, Kenya, **2** Health Economics Research Unit, Kenya Medical Research Institute Wellcome Trust, Nairobi, Kenya, **3** University of Nairobi (UON), Nairobi, Kenya

¤a Current address: USAID-Kenya Integrated Water, Sanitation and Hygiene Project, UN Crescent, Gigiri, Nairobi, Kenya
¤b Current address: Health Economics Research Unit, Kenya Medical Research Institute, Upper Hill, Nairobi, Kenya
¤c Current address: Department of Health Economics, University of Nairobi, Nairobi, Kenya
* dianamulatya@gmail.com

**Data Availability Statement:** All data is contained within the paper or supporting information files.

**Funding:** This work was made possible by the support of the American People through the United

## Abstract

Poor sanitation worldwide leads to an annual loss of approximately $222.9 billion and is the second leading cause of Disability-Adjusted Life Years (DALY's) lost due to diarrhoea. Yet in Kenya, the slow rate and levels at which the household's access improved sanitation facilities remain a concern, and it is unknown if the cost of new technologies is a barrier to access. This study assessed the maximum willingness to pay (WTP) for SAFI and SATO sanitation products and identified those factors that affect the willingness to pay (WTP) valuation estimates by households in three counties in Kenya. It used quantitative economic evaluation research integrated within a cross-sectional survey. Contingent valuation method (CVM) was used to determine the maximum WTP for sanitation in households. We used the logistic regression model in data analysis. A total of 211 households were interviewed in each county, giving a total sample size of 633 households. The mean WTP for SAFI latrines was $153.39 per household, while the mean WTP for SATO pans and SATO stools was $11.49 and $14.77 respectively. For SAFI latrines, households in Kakamega were willing to pay $6.6 more than average while in Siaya, the households were willing to pay $5.1 less than the average. The main determinants of households WTP for the two sanitation products included household's proximity to the toilet (p = 0.0001), household income ($\beta$ = .2245741, p = 0.004), sanitation product ($\beta$ = -2968.091; p = 0.004), socioeconomic status ($\beta$ = -3305.728, p = 0.004) and a household's satisfaction level with the current toilet ($\beta$ = -4570.602; p = 0.0001). Increased proximity of households to the toilet, higher incomes, and providing loan facilities or subsidy to poor households could increase the demand for these sanitation technologies.

States Agency for International Development (USAID) under the terms of Contract No. AID-615-TO-15-00001, formally commissioned as the Kenya Integrated Water, Sanitation and Hygiene (KIWASH) project. The authors received no specific funding for this work as the research was conceived and embedded in ongoing development work, whose project goal was to improve the sanitation status of rural communities in Kenya. The study was therefore developed to provide insights on the demand for products by target users, and inform learning and adaptation of the existing sanitation project. The KIWASH project provided support in the form of salaries for authors [DM,VW] and consultancy fees for author (JO) who worked under a firm, known as Adaptive Management and Research Consultants (AMREC) LTD. The project did not have any additional role in the study design, data collection and analysis, decision to publish, or preparation of the manuscript.

**Competing interests:** This study was undertaken by a Kenyan based consultancy firm, here in referred to as AMREC consultants, and was conducted as part of a consultancy assignment funded by the USAID-KIWASH project. Though the study received no grants for research, we hereby declare competing financial interest from salaries received to support authors to execute the study. Additionally, none of the sanitation technologies assessed belong to USAID/KIWASH project or AMREC. The decision to publish was exclusively made by authors in a bid to contribute to the sanitation sector knowledge. USAID/KIWASH project will neither directly benefit from this publication nor did it play any additional role in the study design, data collection and analysis, decision to publish, or preparation of the manuscript. This engagement does not alter our adherence to PLOS ONE policies on sharing data and materials.

# Background

Access to improved sanitation facilities has several public health benefits, including reduced communicable diseases such as soil-transmitted helminths (STH), cholera, diarrhea, trachoma [1], and malnutrition [2–4]. Other benefits include economic savings [5] due to cut healthcare costs and non- health benefits [6] from time savings otherwise forgone in nurturing patients or queuing to gain access to shared facilities or find privacy for open defecation. Despite known benefits, effective excreta removal remains a challenge, especially in achieving universal coverage for sparsely distributed rural population's characteristics of a majority of residents in low and middle-income countries (LMIC) [7]. Traditional sewers are prohibitively expensive to scale through public finance. Similarly, sewerage business cannot attract private sector investments due to presumably low returns and high-risk conditions. Onsite sanitation, such as ventilated improved pit latrines or latrines connected to septic tanks, are therefore viewed as viable solutions in most rural populations. Whereas onsite facilities present a strong case, they too can be unaffordable to the majority of households in rural areas, whose daily average earning is below a dollar day. Governments in LMIC have limited public funding and are unable or reluctant to finance onsite sanitation against a backlog of other priorities. Most governments in LMIC are still actively engaging stakeholders to expand innovations in affordable and responsive sanitation technologies that meet consumer demand.

In Kenya, the responsibility for access to an improved sanitation facility lies with the individual household, while the Government reserves an overarching role in health education and creating an enabling environment for private sector participation. Only 59.3 percentage of individuals have access to improved sanitation, with the majority of residents using the rudimentary latrine and a staggering 13.9 per cent of rural residents engaging in open defecation [8].

What is worrying in Kenya is not the high number of people without access to safe sanitation facilities but the slow rate at which the households can access improved sanitation facilities. Access to improved sanitation services in Kenya increased by only 1.3 per cent between 2009 and 2015 [9, 10]. Despite the availability of technologies and products in the market, demand for improvements in the quality of sanitation has remained sufficiently low.

Several explanations for poor sanitation include the low priority accorded to sanitation [11], social-economic barriers, colossal capital outlays (without subsidies) and lack of credit arrangements for poor households to invest in sanitation [12, 13], low awareness on health benefits, lack of privacy [14] or availability of infrastructure development such as availability of piped water [15]. Other studies have shown that higher-income households to be positively associated with higher willingness to pay for sanitation, as the number of disposable income increases. Similarly, higher education levels and increased environmental awareness is associated with higher WTP for sanitation improvements [16].

Increasing consumer demand for sanitation in Kenya's rural communities employs proven approaches like Community-Led Total Sanitation (CLTS) [17, 18] and commercial and social marketing principles to trigger positive sanitation practices. These two approaches have gained momentum in the recent past. For instance, a USAID supported Kenya integrated water and sanitation (KIWASH) project is working across nine counties to trigger and activate demand for low–cost affordable sanitation technologies in the country. The project combines Sanitation Marketing and CLTS–an approach that focuses on mobilizing and unleashing the communities' drive to improve their sanitation status and end open defecation without providing material support or subsidies- to achieve its objectives.

Sanitation marketing refers to the use of commercial and social marketing techniques to scale up demand and supply for improved sanitation. In the past, Kenya's sanitation market

was primarily dominated by ventilated improved pit latrines, with partners and Governments promoting manufacture and installation of concrete slabs. Despite massive investments stemming from Governments and donors, these technologies were widely unsuccessful. To increase the range of technological options available in the sanitation market, USAID/KIWASH project partnered with Lixil corporation- SATO product proprietors and other local enterprises to trial and expand supply for SATO and SAFI latrine products respectively. Even though small improvements don't confer optimal sanitation benefits in favor of full latrines, studies [16] have shown that households with demonstrated willingness to make minor home improvements are more likely to switch to using an improved sanitation facility. Partners' efforts focus on ensuring gradual improvements to basic sanitation, with a longer-term trajectory aligned towards improved sanitation systems.

SATO is a simple latrine slab that can be retrofitted on a pit latrine to improve the quality and aesthetics of the structure. On the other hand, SAFI latrine is an improved sanitation facility; typically, a low cost ventilated pit latrine designed to reinforce circular pits against external pressure. Fig 1 in supplemental information shows the two technologies.

Though the project supported demand creation creativities for both products, program results indicated varied uptake between the two products, with SATO products generating higher sales than its counterpart. This study provides answers on the disparate uptake in sanitation technologies by investigating the consumer's WTP at different price levels and assessing underlying drivers for sanitation demand in rural communities.

Findings from this decision-focused evaluation, like other studies [19] with similar designs, will be used to inform the programs to scale up and demand activation strategies. Moreover, understanding the willingness to pay (WTP) for sanitation by customers and the reasons for low WTP in developing countries like Kenya guides effective policy formulation and practice.

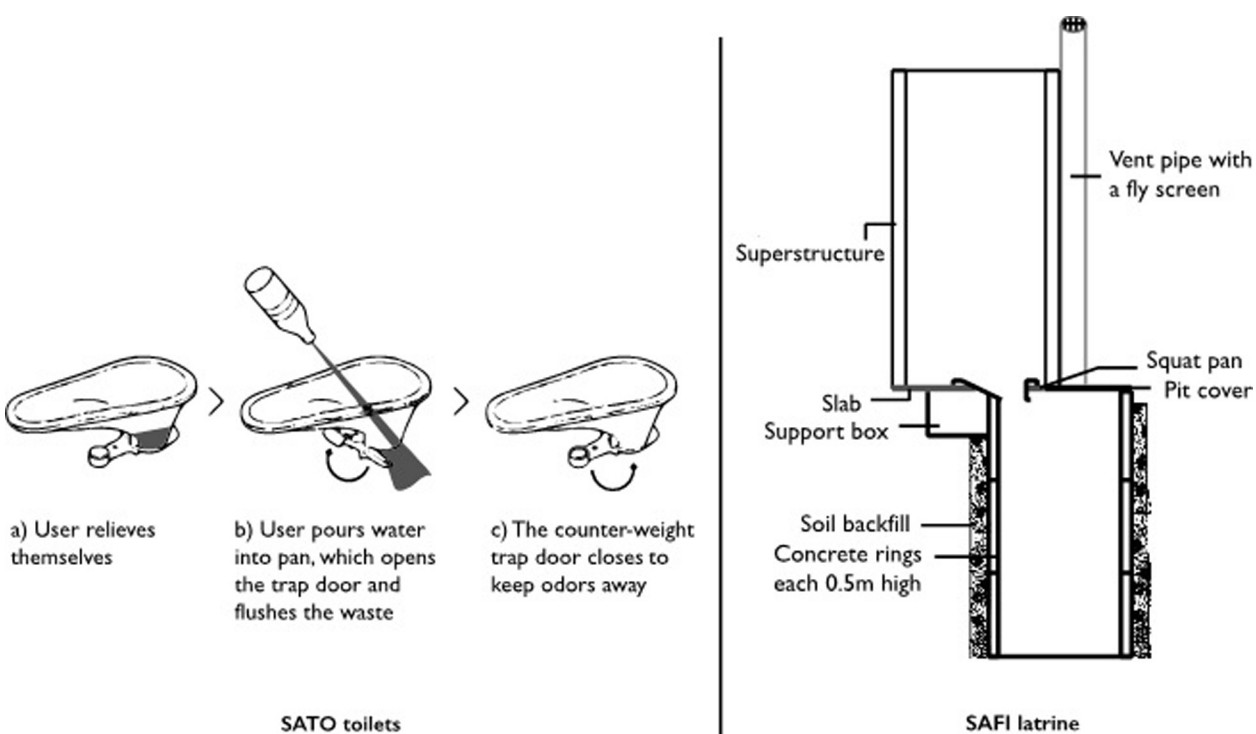

**Fig 1. Cross sectional view of SATO toilets and SAFI latrines.**

The study will assess the households WTP for SAFI latrines, SATO toilet products, and the factors that influence WTP for sanitation in Makueni, Kakamega, and Siaya Counties in Kenya.

## Methods and materials

### Study site

The study was conducted in Siaya, Kakamega, and Makueni counties, which are predominantly rural (about 80%) counties in Nyanza, Western and Eastern regions of Kenya.

Sanitation coverage in rural Kenya is still lacking, with a glaring 50.8% of the population using unimproved sanitation and another 13.9% practicing open defecation rates. In the recent 2015/16 Kenya Integrated Budget Survey report, Siaya, Kakamega, and Makueni reported 56.2%, 67.5%, and 12.1% access to unimproved sanitation facilities, respectively [8]. The three counties have a combined population of 3,572,774 people. The population distribution for Kakamega, Makueni and Siaya is 1,732,145, 921,168 and 919,461 respectively [20]. The study zones were within the three counties (Fig 2), and specifically where project activities centered on improving awareness on the benefits of safe sanitation.

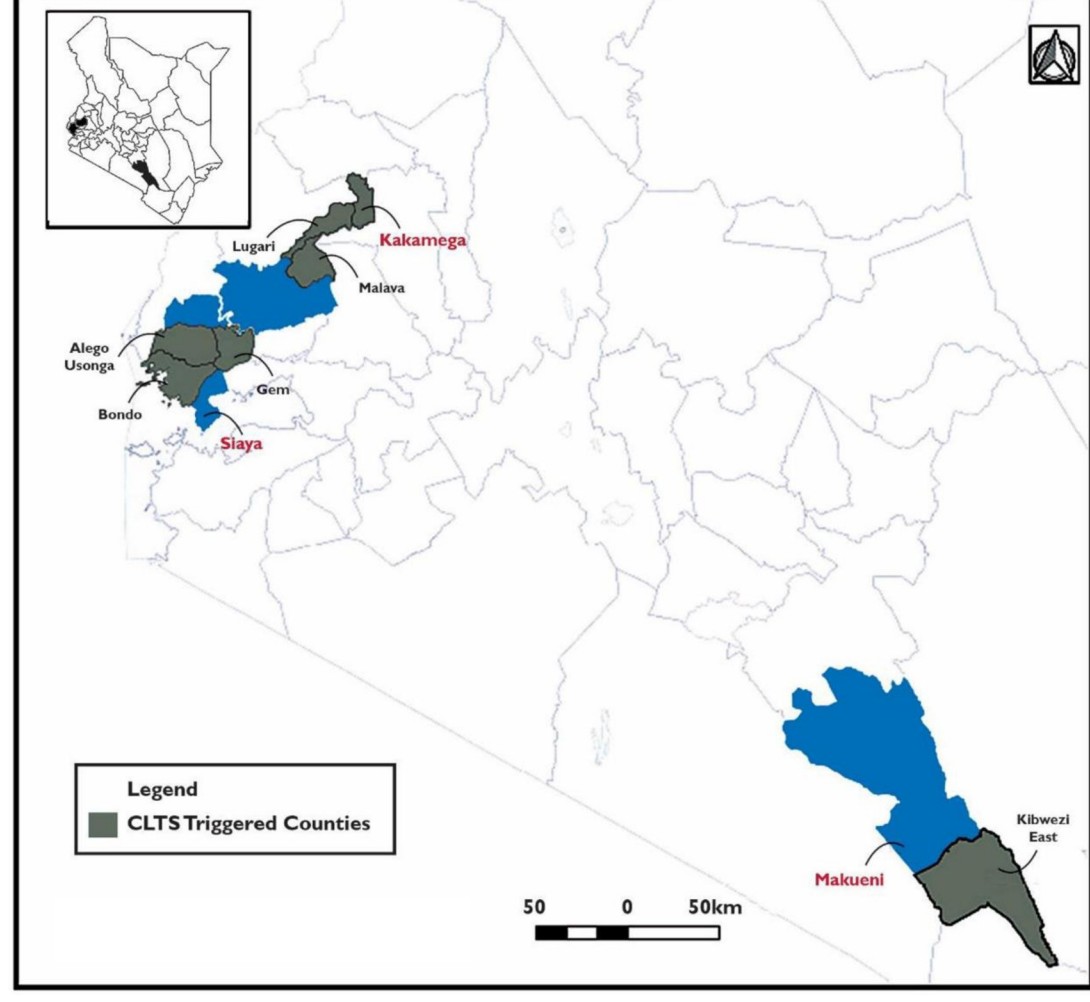

**Fig 2. Map of study areas.**

## Study design

This study was quantitative economic research integrated within a cross-sectional survey, in which a Contingent Valuation Method (CVM) [21] was employed to elicit WTP valuation estimates from households.

CVM is a stated preference method in which respondents are asked their maximum level of willingness to pay for a predetermined increase or decrease in the quantity or quality of a good [21]. It involves asking the respondents a sequence of questions that progressively narrows down to their willingness to pay (WTP). In the dichotomous choice version of CVM, respondents were offered a change in the good at a given cost, and the respondent either accepted or refused payment of the suggested cost [22]. Dichotomous choice method has been shown to generate more robust WTP estimates than open-ended questions which have been shown to produce unrealistic estimates due to the likelihood of getting protest responses from respondents [23, 24]. In this study a bidding game technique that involved a double bounded dichotomous choice with follow up method was used to elicit household WTP valuations. The dichotomous choice with follow-up format did not directly reveal the respondent's WTP; instead it provided a range within which the true WTP lied (double bounded) [23, 25] (i.e., yes or no to the option offered in each question). Before the bidding started, a scenario explanation was sufficiently stated to make the respondent understand the scenario presented to them very well.

## Data collection methods

A questionnaire was developed, upload onto the Open Data Kit software [26], and a team of enumerators trained on data collection. The questionnaire comprised of four sections; section one entailed social-demographic assessment, part two was on the household, and environmental characteristics and part three elicited WTP, using dichotomous choice with a follow-up version of CVM [5] to assess the respondent's maximum WTP for an increase in the quality of sanitation services through purchase and use of SAFI/SATO products. The introductory section had scenario explanation statements about SATO products and SAFI latrine. The respondents were showed colored pictures of each toilet product in turn, followed by a list of dichotomous choice questions (Yes/No) with a bid attached to each question. To minimize the starting point bias, two versions of the CVM questionnaire were randomly [13] administered to the respondents. The subsequent section assessed the role of credit on the choice of WTP valuations. The CVM scenario was repeated to the respondent but with an offer of constructing a SAFI latrine on a 10% interest loan repayable within 12 months. Pre-survey, the questionnaire was pretested by trained enumerators and the feedback used to improve the reliability of the instrument. A total of 553 households were interviewed in the survey.

## Sampling strategy

The master sampling frame consisted of villages in Siaya, Kakamega, and Makueni, where the project was working to promote awareness on benefits to improved sanitation.

The sampling process entailed a two-stage process. A simple random sampling technique was employed to select the number of villages. Systematic random sampling was used to identify target households selected from these villages. Only households without SATO products or SAFI toilet products were included in the study. Enumerators were guided on eligible households for interviews by community health volunteers (CHV's). These CHV's had worked with the county Ministry of health staff in implementing water, hygiene, and sanitation interventions.

## Data analysis

A Tobit regression model was used to estimate factors that influenced the maximum amount of households' WTP for sanitation within the study counties. Tobit regression models have been widely used by economists to identify the determinants of willingness to pay for various products, including sanitation products.[27–29] A Tobit regression model is a type of a limited dependent variable model in which there is a censored sample, a sample in which the information of the regressand is available for some observations but not although there may be information for all the regressors in the sample [30]. The regressand can be left-censored (cannot take a value below a certain threshold) or right-censored (cannot take a value above a certain threshold), or it can be both left and right-censored. In these analyses, the dependent variable is binary (whether willing to pay or not). The marginal effects of the factors were assessed at statistical significance levels of $p<0.05$. Tobit regression is p Referred over other OLS models because of its ability to take care of households with less than zero WTP responses [31] and provide robust and consistent model parameters.

In this study, Tobit regression was chosen because many households could not state the willingness to pay a price less than zero since they were not given a chance to report such WTP responses in the data collection tool. The General empirical Tobit regression model [31, 32] is given by:

$$WTP_i = \beta_i X_i + \varepsilon_i \tag{1}$$

Where;

WTP is the willingness to pay estimate for the ith individual, $\beta$ is a vector of model parameters, X is the vector of explanatory factors, and $\varepsilon i$ is Error term. The following is the linear functional relationship between the household WTP is given by:

$$WTP = \beta_0 + \beta_1 X + \beta_2 X_2 + \ldots\ldots\ldots + \beta_n X_n + \varepsilon \tag{2}$$

Where; X represents determinants of household Willingness to pay (WTP), and $\beta$ is a vector of the coefficients of socioeconomic determinants of household WTP.

Households were further ranked in wealth quintiles (poorest, second poor, middle poor, fourth less poor, and least poor) using principal component analysis (PCA) using household assets, utilities, and characteristics. These variables were included in the PCA model, and the sum of individual weights generated a wealth index for each household. The larger the weight, the wealthier a household is compared to the rest in the sample.

We further assessed the socioeconomic inequities in the distribution of household proportion willing to pay for the SAFI latrines using the Gini coefficient. The Gini coefficient can vary from 0 (perfect equality, also represented as 0%) to 1 (perfect inequality, also represented as 100%). A Gini coefficient of zero means that everyone has the same outcome. In contrast, a coefficient of 1 represents a single individual receiving all the outcome (of course, neither of these extremes are very likely). The higher the Gini index, the greater the degree of inequality; in the case of a very high Gini index, those with high outcomes are disproportionately large in total outcome or events.

The Gini coefficient is equal to the area between the actual income distribution curve and the perfect income equality line, scaled to a number between 0 and 100. The Gini coefficient is the Gini index expressed as a number between 0 and 1 [33].

## Ethics approval

We obtained ethics approval from the Kenya Medical Research Institute. Research participants provided written consent to participate in surveys.

## Results

### Demographic characteristics of the study population

A total of 633 household heads were sampled, out of which 553 were successfully interviewed in the survey, representing a response rate of 87.2%. Table 1 below, shows a breakdown of all households (HH's) that were interviewed in Kakamega (187), Makueni (172) and Siaya (194) County and the demographic characteristics of the study population. Most of the respondents were females 383 (69.3%), while males were 170 (30.7%). Kakamega 136 (72.7%) and Siaya 135 (69.6%) counties had the highest number of female respondents.

Ages of most household heads ranged from 30 years to 59 years, and most of them were married 429 (77.5%). The mean age household heads were 31.9 years (SD+_15.5), and the median age was 31 years. The majority of the household heads had attained primary level education 306 (55.3%). In contrast, only 9 (1.63%) had a university-level education. Majority of the households had between one to five members 333 (60.2%). Of the 54 households with a person with diarrhea (9.78%), most (6.51%) were persons aged >5 years. Majority of households had their toilets less than 20 metres from the house (n = 331; 59.86%), and a majority (n = 378; 68.35%) of them were using improved toilet facilities. Out of 378 households who were using improved toilets, 137 (36.4%) households were in Makueni, 89 (23.5%) in Siaya, and 152 (40.2%) in Kakamega. Out of 553 households interviewed, 209 (37.79%) were satisfied with their current toilets, while 135 (24.4%) households were not satisfied. The majority of households 332 (60.04%) were poor, while 221 (39.96%) were the least poor households.

### Willingness to pay for sanitation

**Mean willingness to pay by sanitation technology.** Table 2 highlights the demand for sanitation technologies in our study population. The overall maximum WTP estimate for SAFI latrine for the pooled data was KES. 15,398 (USD153.98) per household. The mean WTP for SAFI latrine is $147 lower than the average market price of a complete system at KES. 30,000 (USD300). The mean WTP for the installation of a SATO pan and a SATO stool was KES. 1,148 ($11.48) and KES. 1,475.9 ($14.759) respectively.

From Fig 3, only 3% of the sampled households are willing to pay for SAFI latrines at a market price of $300. The proportion of households willing to pay for a SAFI latrine reduces as the price increases in line with the theory of demand and supply. In contrast, demand for SATO products increases as the price increases. The WTP for a SATO pan also follows a similar trend as a SATO stool. Out of 553 interviewed households, 76.2% of them were willing to pay for SATO pans at rates slightly above the retail market price of $ 8 per unit.

Similarly, 78% were willing to pay for SATO stools at rates above the retail market price of $10 per unit. Mean WTP for SATO products varies in wealth quantiles.

However, there is no significant difference in WTP estimates for SATO products when prices are compared among the three study Counties.

**Mean willingness to pay for SAFI latrines.** Fig 4 (a-c) shows the patterns of WTP for SAFI latrines in the three counties. The mean WTP for SAFI latrines in Kakamega and Makueni was almost similar, though well below the estimated market price of KES. 30,000 (USD300). In Makueni, the median WTP was approximately 50% ($ 100) lower than the mean WTP ($150) observed in Siaya and Kakamega counties. However, when WTP bids increased beyond KES. 25,000 ($250) as more households in Siaya and Makueni are now willing to pay higher compared earlier prices.

**Socio-economic profile of SAFI latrine adopters.** Fig 5 below is a Lorenz curve showing the distribution of the household population willing to pay for SAFI latrines by its wealth

**Table 1. Socio-economic characteristics in the study population (values in the parentheses are column percentages).**

| Characteristics | Kakamega | Makueni | Siaya | All Counties |
|---|---|---|---|---|
| **Gender** | | | | |
| Male | 51 (27.27) | 60 (34.88) | 59 (30.41) | 170 (30.74) |
| Female | 136 (72.73) | 112 (65.12) | 135 (69.59) | 383 (69.26) |
| **Marital Status** | | | | |
| Divorced | 2 (1.07) | 2 (1.16) | 0 (0.0) | 4 (0.72) |
| Married | 155 (82.89) | 144 (83.72) | 130 (67.01) | 429 (77.58) |
| Separated | 0 (0.0) | 1 (0.58) | 2 (1.03) | 3 (0.54) |
| Single | 10 (5.35) | 5 (2.91) | 5 (2.58) | 20 (3.62) |
| Widow/widower | 20 (10.70) | 20 (11.63) | 57 (29.38) | 97 (17.54) |
| **Age** | | | | |
| 15–29 | 20 (10.70) | 8 (4.65) | 10 (5.15) | 38 (6.87) |
| 30–44 | 56 (29.95) | 30 (17.44) | 56 (28.87) | 194 (35.08) |
| 45–59 | 69 (36.90) | 69 (40.12) | 56 (28.87) | 194 (35.08) |
| 60–74 | 28 (14.97) | 50 (29.07) | 61 (31.44) | 139 (25.14) |
| 74+ | 14 (7.49) | 15 (8.72) | 11 (5.67) | 40 (7.23) |
| **Education level** | | | | |
| College/Mid-level | 26 (13.90) | 3 (1.74) | 7 (3.61) | 36 (6.51) |
| Other | 21 (11.23) | 20 (11.63) | 23 (11.86) | 64 (11.57) |
| Post-primary/vocational | 7 (3.74) | 10 (5.81) | 1 (0.52) | 18 (3.33) |
| Primary | 84 (44.92) | 109 (63.37) | 113 (58.25) | 306 (55.33) |
| Secondary/A-level | 45 (24.06) | 29 (16.86) | 46 (23.71) | 120 (21.70) |
| University | 4 (2.14) | 1 (0.58) | 4 (2.06) | 9 (1.63) |
| **Occupation** | | | | |
| Formal | 14 (46.67) | 7 (23.3) | 9 (30.0) | 30 (5.42) |
| Casual | 143 (33.97) | 128 (30.4) | 150 (35.63) | 421 (76.13) |
| Unemployed | 4 (26.67) | 2 (13.3) | 9 (60.0) | 15 (2.71) |
| Other | 26 (29.89) | 35 (40.23) | 26 (29.89) | 87 (15.73) |
| **Monthly Income** | | | | |
| <10001 | 174 (93.05) | 154 (89.53) | 175 (90.21) | 503 (90.96) |
| 10001–20000 | 8 (4.28) | 13 (7.56) | 13 (6.70) | 34 (6.15) |
| 20001–30000 | 3 (1.60) | 2 (1.16) | 6 (3.09) | 11 (1.99) |
| 30001–40000 | 1 (0.53) | 0 (0.00) | 0 (0.00) | 1 (0.18) |
| >40001 | 1 (0.53) | 3 (1.74) | 0 (0.00) | 4 (0.72) |
| **Household Size** | | | | |
| 1–5 | 106 (56.68) | 100 (58.14) | 127 (65.46) | 333 (60.22) |
| 6–10 | 7 (3.74) | 3 (1.74) | 3 (1.74) | 3 (1.55) |
| 11+ | 74 (39.57) | 69 (40.12) | 64 (32.99) | 207 (37.43) |
| **Having a person with diarrhoea** | | | | |
| <5yrs | 9 (4.81) | 1 (0.58) | 8 (4.12) | 18 (3.25) |
| >5yrs | 14 (7.49) | 4 (2.33) | 18 (9.28) | 36 (6.51) |
| **Distance to the current toilet** | | | | |
| <20m | 132 (70.59) | 82 (47.67) | 117 (60.31) | 331 (59.86) |
| 21-40m | 44 (23.53) | 17 (9.88) | 44 (22.68) | 105 (18.99) |
| 41-60m | 6 (3.21) | 70 (40.70) | 16 (8.25) | 92 (16.64) |
| 61-80m | 3 (1.60) | 1 (0.58) | 1 (0.52) | 5 (0.90) |
| >80m | 2 (1.07) | 0 (0.00) | 1 (0.52) | 3 (0.54) |
| **Satisfaction with current toilet** | | | | |

*(Continued)*

**Table 1.** (Continued)

| Characteristics | Kakamega | Makueni | Siaya | All Counties |
|---|---|---|---|---|
| Not satisfied | 31 (16.58) | 40 (23.26) | 64 (32.99) | 135 (24.41) |
| Satisfied | 106 (56.68) | 35 (20.35) | 68 (35.05) | 209 (37.79) |
| Moderately satisfied | 39 (20.86) | 92 (53.49) | 57 (29.38) | 188 (34.00) |
| Very satisfied | 11 (5.88) | 5 (2.91) | 5 (2.58) | 21 (3.80) |
| **Main Water Sources** | | | | |
| Improved | 134 (71.66) | 55 (31.98) | 49 (25.26) | 238 (43.04) |
| Unimproved | 53 (28.34) | 117 (68.02) | 145 (74.74) | 315 (56.96) |
| **Toilet facility** | | | | |
| Improved | 137 (73.26) | 89 (51.74) | 152 (78.35) | 378 (68.35) |
| Unimproved | 50 (26.74) | 83 (48.26) | 42 (21.65) | 175 (31.65) |
| **Having children <5yrs** | | | | |
| Yes | 108 (57.75) | 67 (38.95) | 64 (32.99) | 239 (0.37) |
| No | 79 (42.25) | 105 (61.05) | 87 (44.85) | 271 (0.42) |
| **Economic Status** | | | | |
| Poor (Quntile1-3) | 159 (85.03) | 34 (19.77) | 139 (71.65) | 332 (60.04) |
| Least Poor (Quintile 4–5) | 28 (14.97) | 138 (80.23) | 55 (28.35) | 221 (39.96) |
| **Taken credit previously** | | | | |
| Yes | 49 (26.2) | 44 (25.6) | 86 (44.3) | 179 (32.4) |
| No | 138 (73.8) | 128 (74.4) | 108 (55.7) | 374 (67.6) |

quintiles. The area between the curve and line of equality represents the level of inequality between the poorest and least poor households. The Gini coefficient is 0.238 amongst those willing to pay for SAFI compared to 0.391 amongst those unwilling to pay for SAFI. A positive value represents a largely rich distribution in WTP bids for SAFI latrines. It is also worth noting that, the inequality gap is narrower for households willing that those unwilling to pay for sanitation.

Fig 6 shows that inequalities were least evident in Kakamega with (Gin = 0.196) compared to Makueni (Gini = 0.0255) and Siaya (Gini = 0.241) Counties. Richer households in Makueni dominated the proportion of households willing to pay for SAFI. In semblance to Fig 5, richer households still dominate amongst those willing to pay for SAFI latrines in all counties.

**Factors influencing willingness to pay for sanitation products.** From Table 3 below, it can be inferred that distance to the current toilet (p<0.0001), current use of unimproved toilet (p = 0.0001), being a resident of Makueni and Siaya County (p = 0.00, p = 0.033), household income (p = 0.005), being moderately satisfied with current toilet (p = 0.0001) and being a poor household (p = 0.005) determine the amount of money that households are willing to pay for the construction of a SAFI latrine. Factors that would increase the probability of a household's WTP for a SAFI latrine were those with at least a 5% level of significance and those with a positive coefficient. Those most likely to purchase SAFI latrines were households that had to walk longer distances to the toilet and those with increased household incomes. Similarly, a

**Table 2. Mean estimates for willingness to pay for sanitation in the study population.**

| Sanitation Type | Poorest | Wealthiest | Mean | Market Price |
|---|---|---|---|---|
| Safi | 13,527 ($135.28) | 18,148 ($181) | 15,359 ($153.59) | 30,000 ($300) |
| Sato pan | 1,083 ($10.83) | 1,248 ($12.49) | 1,148 ($11.49) | 800 ($8) |
| Sato stool | 1,390 ($13.91) | 1,608 ($16.09) | 1,476 ($14.77) | 1,000 ($10) |

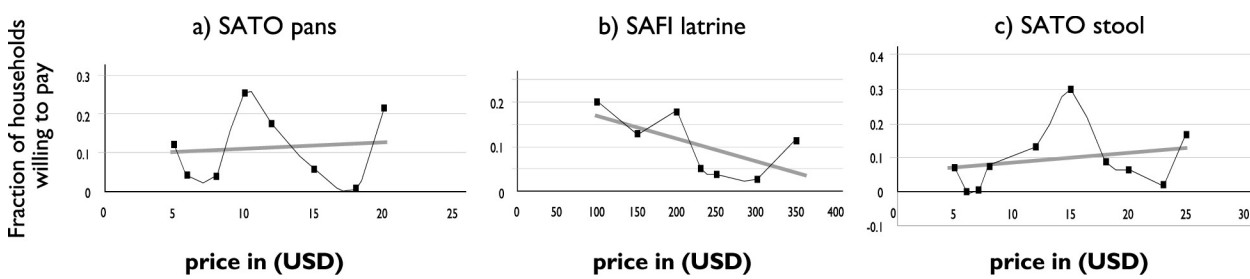

**Fig 3.** (a-c). Demand curves for SATO toilet products and SAFI latrines.

unit increase in monthly income increases the maximum amount of money that the household is willing to pay, by KES. 0.224, i.e., an increase in monthly household income of KES. 1,000 (10 US$) increases the maximum amount that the household is willing to pay by KES. 224 (2.24 US$) per month. This positive relationship is in concurs with other similar studies [15, 16, 34]. Other factors associated with a reduced probability of purchasing SAFI latrines, were those with a negative coefficient and included being a resident of Makueni and Siaya compared to Kakamega, households in the poorest wealth quintiles, and if current households used unimproved facilities.

Further results from Table 3 confirm that an increase in the distance to the current toilet (p = 0.0001), dissatisfaction with the current sanitation conditions (p = 0.029), and higher household incomes (β = 0.0113098; p = 0.011) increased overall willingness to purchase SATO pans. On the other hand, the willingness to pay for a SATO stool is influenced by the distance to the current toilet (β = 8.604; p = 0.0001), use of unimproved toilet (β = -163.800 p = 0.024), being a male household of the head (β = 135.04; p = 0.039) and satisfaction with the current toilet facility (β = -162.822, p = 0.0030; β = -341.663, p = 0.0001).

## Influence of finance on sanitation uptake

Fig 7 below compares the maximum WTP price for construction of SAFI Latrines when households were asked to pay cash up front and when given credit finance. Overall, credit has a positive impact on households WTP estimates; the household's mean WTP valuations changed by 2.3%, from KES.15, 398 (USD$ 153.98) with cash upfront payments to KES.15, 722N(USD$157.22) on credit terms. A higher proportion of households (19%) in Makueni were willing to purchase SAFI latrines if credit was offered compared to cash term payment. In contrast, households in Kakamega and Siaya counties were unwilling to pay for SAFI latrines, with a -4% and -1.6% reduced WTP, respectively (Fig 7). Overall, there is a

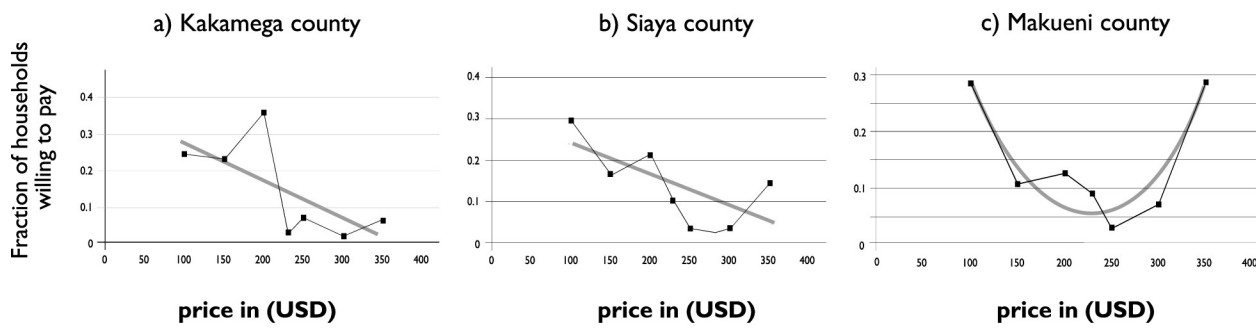

**Fig 4.** a-c. Demand curves for SAFI latrines by county location.

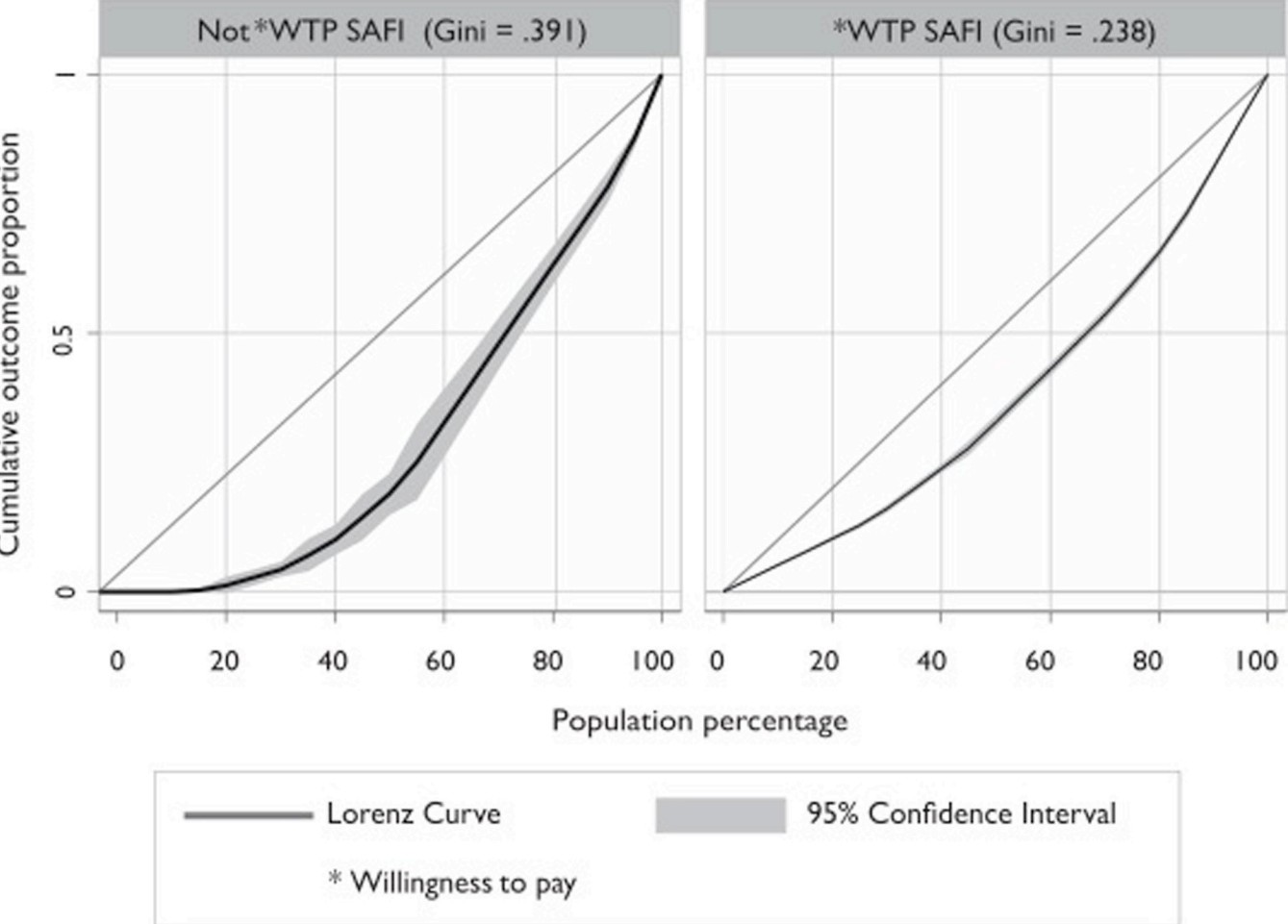

**Fig 5. Lorenz curves comparing inequality ratios amongst households in the study population.**

marginal difference between the households that were willing to pay for a SAFI latrine using a loan and without a loan.

Fig 8 shows that households without SAFI latrines were willing to pay cash upfront up to a maximum price of 220–230 US$, beyond which demand for credit financing increases slightly. However, similar slopes observed in demand curves for households with/without credit financing imply that the availability of credit finance does not highly influence demand for SAFI latrines.

## Discussion

From Fig 3, the percentage of households WTP for SAFI latrines declines with increasing price bids, with the majority of HH's (20%) willing to pay a meager $100, while only 3% are willing to pay SAFI latrines at the recommended market price of $300. This results are in congruence with other studies[35, 36] that confirmed that demand for health-related products is price elastic. Conversely, demand for SATO pans and stools increased with price increase up to a maximum value of approximately KES.1, 148 ($11.48), and KES.1, 475 ($14.75), respectively

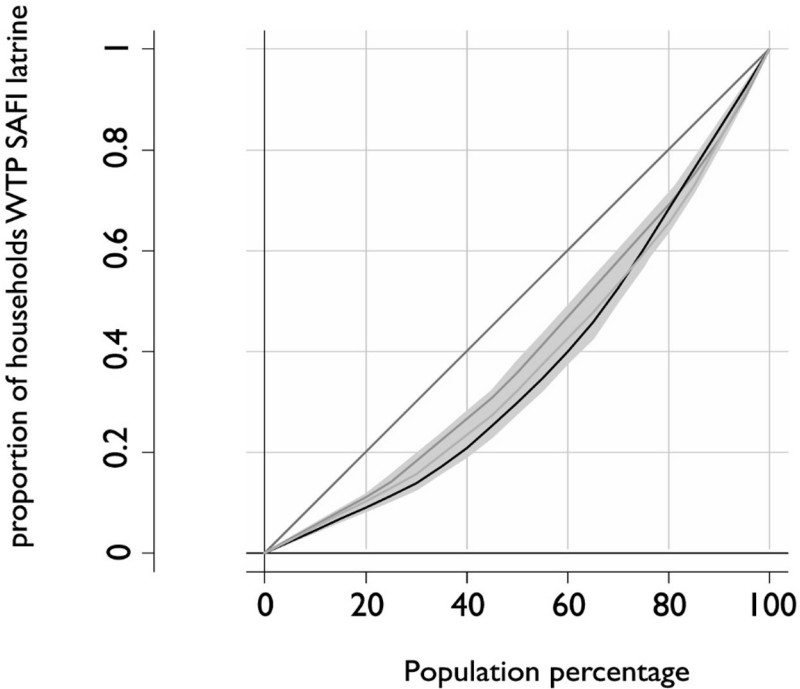

Fig 6. Lorenz curves showing wealth inequality for SAFI in study counties.

beyond which it rapidly declined (Table 2). Findings (Fig 3) show that more than half of study households are willing to pay for SATO pans and stools above the recommended market retail price of KES. 600 (6US$) and KES. 1000 (10US$), respectively. This results confirm a high demand for SATO products at the households and are quite similar to another study conducted in Ethiopia; [37]. This study reported that the product aesthetics, fair price, ability to reduce smell, and lock in flies were all responsible for high demand for SATO products in rural households. Therefore, results from these two studies predict better business prospects for traders in SATO products.

In contrast, these findings were not comparable with a recent Kenyan study [38] that reported less than 4% of households as willing to pay for plastic slabs. Though plastic slabs are considered to be close products to SATO toilets, they are defined as a type of a different retrofit latrine slab that is less attractive and higher priced when compared to SATO. Further social science investigations could reveal the reasons for the preference observed in SATO products and insights used by private sector to better innovate along product attributes highly prefered by the local sanitation users.

As shown in Table 2, the overall mean WTP for the construction of a model SAFI latrines was KES.15, 398 ($153.98) and Fig 3 shows that the actual construction cost of a SAFI latrine, approximated at (300US$), was way above the price valuation expressed by many households.

**Table 3. Summary multivariate results on the determinants of WTP for sanitation products in households.**

| Variable | SAFI | | SATO pan | | SATO stool | |
|---|---|---|---|---|---|---|
| | Coeff | P-value | Coeff. | P-value | Coeff. | P-value |
| Distance to the toilet | 152.27 | 0.000* | 6.204242 | 0.000* | 8.604276 | 0.000* |
| **County** | | | | | | |
| Kakamega (Ref) | | | | | | |
| Makueni | -3975.152 | 0.007* | -119.9807 | 0.147 | -50.5208 | 0.623 |
| Siaya | -2529.206 | 0.033* | -154.9121 | 0.018** | -266.885 | 0.001* |
| **Satisfaction** | | | | | | |
| Not satisfied (Ref) | | | | | | |
| Moderately | -4696. 026 | 0.000* | -310.7425 | 0.000* | -341.663 | 0.000** |
| Satisfied | -2081.854 | 0.052 | -131.5552 | 0.029* | -162.822 | 0.030** |
| Very satisfied | -2101.279 | 0.378 | -29.88273 | 0.822 | -154.615 | 0.350 |
| **SES** | | | | | | |
| Least Poor (Ref) | | | | | | |
| Poor | -3264.145 | 0.005* | -74.52313 | 0.254 | -81.1948 | 0.318 |
| **Gender** | | | | | | |
| Female (Ref) | | | | | | |
| Male | 1536.69 | 0.124 | 30.25767 | 0.565 | 135.0432 | 0.039** |
| **Sanitation** | | | | | | |
| Improved (Ref) | | | | | | |
| Unimproved | -3075.035 | 0.003* | -95.45451 | 0.100 | -163.800 | 0.024** |
| **Income** | 0.2236298 | 0.005* | 0.0113098 | 0.011* | -0.000199 | 0.971 |

* = P<0.05 are significant determinants of household WTP valuations estimates.

Ref = Reference category.

Coeff. = coefficient.

Moreover, the Lorenz curves (Fig 6) showed that households from higher socioeconomic groups dominated the willingness to pay for SAFI latrines across all study counties. These findings emphasizes the need for Governments to design pro-poor subsidies that specifically reach the bottom of pyramid sanitation users, who an unable to pay for improved sanitation products.

Fig 4 showed that residents of Makueni County were associated with a lower WTP for SAFI latrines, i.e., residents of Makueni were less likely to purchase the products compared to those in Siaya and Kakamega. The scores may be attributed to a higher level of poverty in the sampled villages in Makueni. Similar differences were observed in other studies [39] in Kenya, where different geographical locations expressed varying willingness to pay for the same sanitation services.

From Fig 7, more households (19%) in Makueni County would purchase the SAFI latrines if given credit while in stark contrast, the willingness to pay for SAFI latrines decreases when households in Kakamega and Siaya counties are offered credit financing. Past studies [12, 40–42] have confirmed that providing a credit or loan facility to rural households can be a critical driver of demand for these products, especially the construction of a SAFI latrine, which requires a substantial amount of money that is not easy to get in the rural areas. Several past evaluations [43–45] of microfinance have suggested that the poorest people benefit the least as most are too scared of failing to comply with regulations requiring short repayment periods or are just unwilling to commit to group obligations and ensuing debt. Though most available literature is coherent with Makueni results, Kakamega and Siaya households seemed to have a

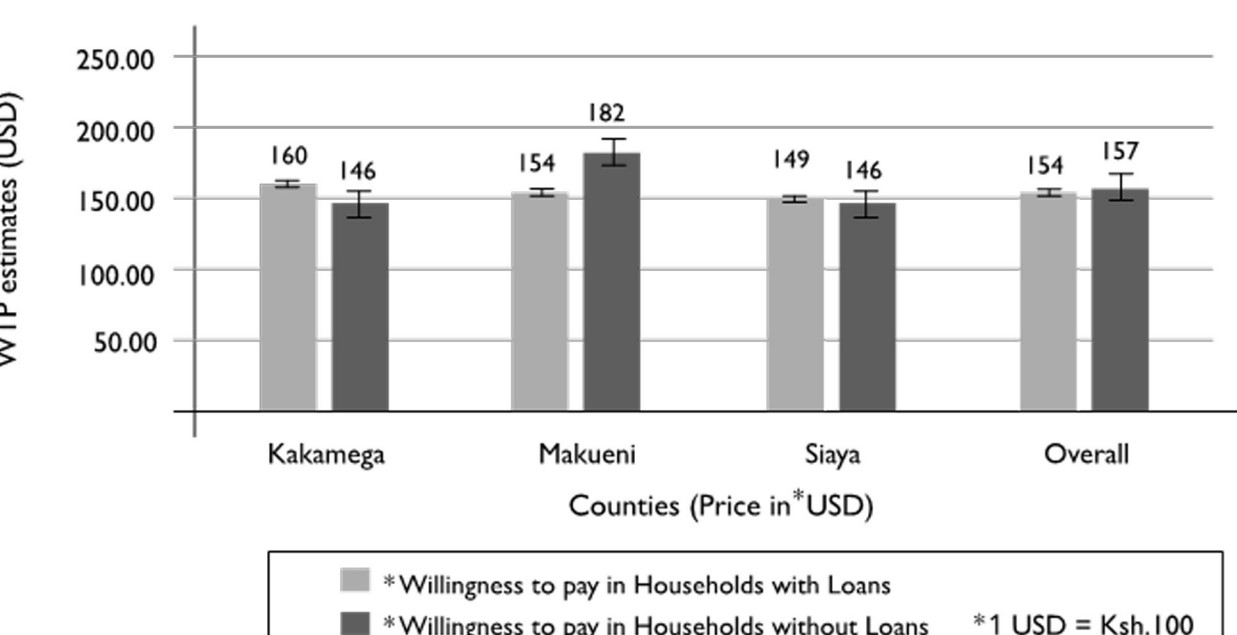

**Fig 7. Mean Willingness to pay for SAFI latrines amongst households with/without finance.**

dissenting attitude towards borrowing, an observation that could be better understood with further social science investigations.

We speculate that past bitter experiences with micro-finance institutions operating in Siaya and Kakamega may have deemed micro-finance loans as untrustworthy, risky, and, therefore, not a cost-efficient strategy for financing sanitation investments. It's also worth noting that historically, Kenya's banking sector has also been accused of cartels for harassment and fraudulent repossession of collaterals issued by clients seeking loans [46, 47]. Alternatively, using a different perspective that is primed on utilizing trusted financing vehicles, such as the village savings and loan associations (VSLA's) may be a more successful strategy for offering credit finance opportunities to Kenya's rural residents. Our views support VSLA's concept, mainly because Kenya's informal finance is estimated to operate at 15.1% in the lowest wealth quantiles and one in every three women is a member of a local community savings and credit group [46]. These results suggest the need for a better understanding of factors that affect acceptability of financial interventions in communities.

Table 3 shows that the demand for our sanitation products is influenced by several factors, one amongst them was the distance from households to the current toilet (p = 0.0001). For every unit increase in distance (meters) from household to the toilet, the maximum amount of money that the household is willing to pay increases, by KES 152.3 (US$1.523). Therefore, the shorter the distance between the toilet and the house, the less likely a household will purchase the products. In rural Kenya, communities tend to share communal latrines, where the distance from individual homes to latrines is usually a deterrent to use, especially during the night. Therefore, residents must look for alternative ways to meet their needs as access to these facilities can particularly expose women and children to heightened risks [45, 46, 48]. Our findings are similar to other studies [49], which found that households are willing to pay for sanitation services when the households lacked an indoor toilet.

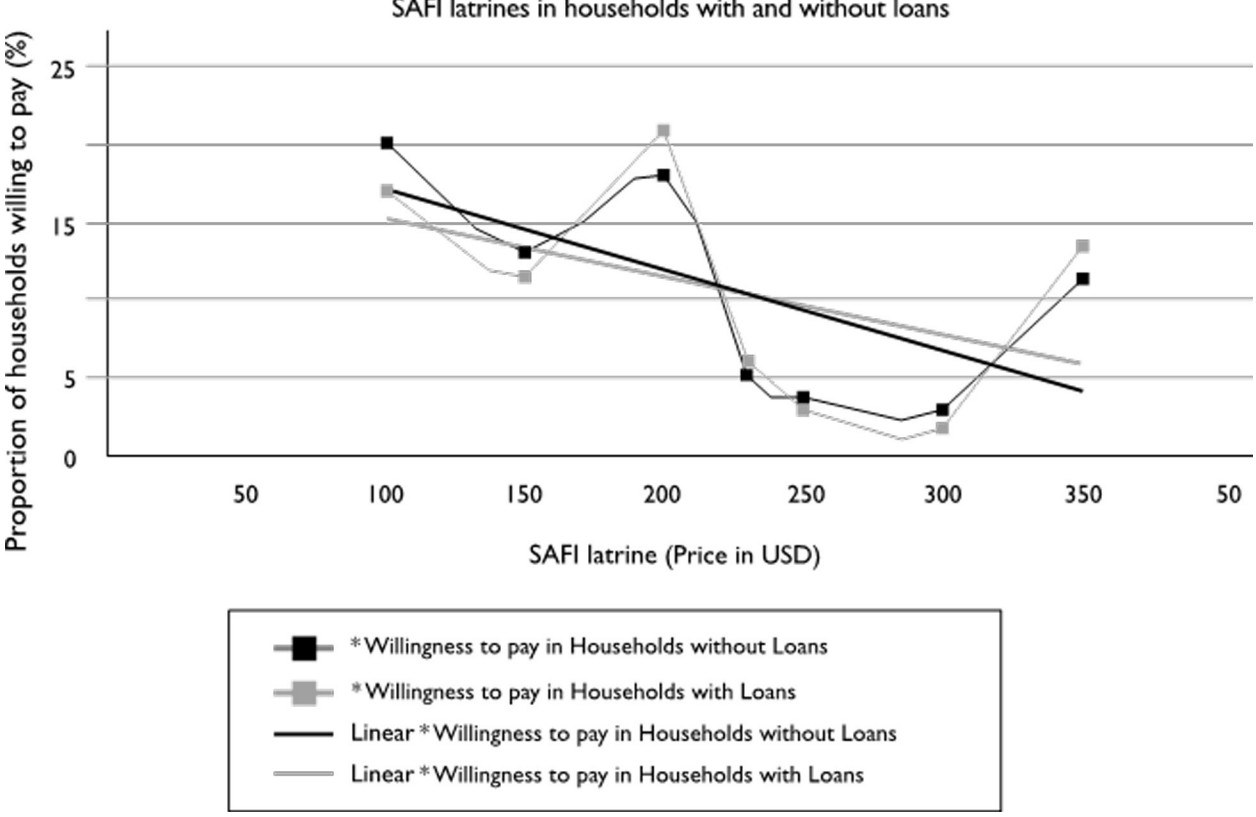

**Fig 8. Demand curves for SAFI latrine in households with/without financing.**

From Table 3, households satisfied with the state of the current toilet facility (β = -162.822, p = 0.03), and those using unimproved toilet facilities (β = -163.8, p = 0.024) are unlikely to purchase SATO products. We speculate that this could be due to a lack of confidence with the current sanitation facilities, previous disappointments with some type of facilities, or even lack of awareness on the benefits of the new products. Those households who were more satisfied may have derived sanitation benefits and may be more willing to purchase or try new products. Our argument contrasts a study conducted in Peru that reported that the lower the satisfaction with sewered sanitation service, the higher the WTP [49]. Other in-depth studies [19, 50, 51] have shown that increased awareness and motivation can strongly have a positive shift on the willingness to purchase sanitation technologies.

A households wealth quantile (β = -326.4, p = 0.005) and income (β = 0.2236, p = 0.005) influenced rural households' WTP valuation estimates for a SAFI latrine, while income (β = 00113, p = 0.011) influenced WTP for SATO pans. Both variables, poverty level and income did not affect WTP valuation estimates for SATO stools. We speculate that households in the poorer wealth quintiles are unlikely to purchase SAFI latrine due to either lack of adequate income, low priority in terms of needs, the vulnerability of such households, and feeling of social exclusion. Studies [16, 34] have shown that the poorer the household is, the less likely they will access and use sanitation products, and that these households are often at risk of adverse events or outcomes. A study by Bosh et al. [34] established that inadequate water and sanitation services to the poor increase their living costs, lower their income earning potential, damage their well-being, and make life riskier.

### Limitations of the study

First, our study was based on stated preferences expressed from interviews but not revealed data through market data. Though the study findings are similar to observed uptake of the two technologies under the project, stated preferences may differ significantly from actual p References, demand and behavior. Other studies using stated preference surveys give respondents time or opportunities to purchase products, which can be advantageous. Due to limited resources and the timeline for the project, the survey respondents were not given time to give options or reveal demand, though interviewers took time to explain the products and simulate the market scenarios in all cases.

Secondly, our work in promoting sanitation coverage through the construction of Sato toilets and SAFI latrine in rural areas was specific to our interventions and the local context, and therefore may not be generalizable to other sanitation technologies in other settings.

## Conclusion

The survey revealed that prices were not a determinant for demand in SATO products in the three counties. To increase sanitation coverage with SATO products, we recommend use of incentives that motivate front line demand activators and marketing agents capable of activating the latent demand observed in communities. SATO product manufacturers, Government or donors should also prioritize investments in educating and building communities awareness on the product availability and accessibility mechanisms established in all study counties. Interestingly, results in Makueni imply that more households are willing and comfortable to pay for SATO products than the other two counties.

This may be a combination of other factors, accessibility of products and increased awareness on product attributes owing to the massive promotional campaigns rolled out in Makueni County. From results, the willingness to pay for the SAFI latrines was inequitable and only affordable for those in higher socioeconomic quintiles. We recommend evaluating the impact of promotional and marketing strategies used for each county, where insights drawn from the study can be used to inform adaptations on project implementation. Such adaptations could include applying an appropriate subsidy targeting poor households to promote access to onsite sanitation. Alternatively, Governments and project implementers could facilitate engagement with sanitation producers or manufacturers, facilitating an enabling environment that crowds-in potential private investors in the sanitation sector. For instance, SAFI latrine producers should be encouraged to re-think and design innovations that reduce production cost for SAFI latrines and ultimately increase affordability to the majority of households. Our results on the impact of finance on WTP for SAFI latrines show divergent and conflicting results. This only confirms that a one size fits all sanitation strategy for Kenya is bound to fail.

Across all study counties, factors that significantly and positively influenced the WTP for SAFI latrines included; increased distance from households to the toilet, household incomes, and being residents of Kakamega county. A household with high poverty levels had a negative relationship on willingness to pay. Increased household income positively influenced the WTP for SATO pans, but not the SATO stool. Satisfaction with current sanitation facilities was associated with high WTP, while the current use of unimproved toilets was associated with reduced WTP for all sanitation technologies. This study highlights the underlying barriers and drivers for demand in the two sanitation technologies and provides a basis for developing policies that support further research and development of sanitation prototypes in Kenya.

Our findings highlight the need for development of flexible and guiding sanitation policies rather than embracing rigid and prescriptive guidance on technologies for going to scale with

rural sanitation programs. From the survey, we also demonstrate how the use of knowledge from program implementation is combined with surveys to guide changes in implementation and promote program efficiency improvements over a projects life span.

## Supporting information

**S1 Appendix. Survey tools administered to respondents.**
(PDF)

**S1 Dataset. Survey data.**
(CSV)

## Acknowledgments

The contents of this article are the sole responsibility of the authors and do not necessarily reflect the views of USAID or the United States Government. Use of the content for non-commercial use is authorized, provided the source is acknowledged.

## Author Contributions

**Conceptualization:** Diana Mutuku Mulatya, Japheth Mbuvi.

**Data curation:** Diana Mutuku Mulatya, Vincent Were.

**Formal analysis:** Vincent Were, Joseph Olewe.

**Funding acquisition:** Japheth Mbuvi.

**Investigation:** Diana Mutuku Mulatya, Vincent Were, Joseph Olewe.

**Methodology:** Diana Mutuku Mulatya, Joseph Olewe.

**Project administration:** Diana Mutuku Mulatya, Japheth Mbuvi.

**Resources:** Diana Mutuku Mulatya, Japheth Mbuvi.

**Software:** Vincent Were, Joseph Olewe.

**Supervision:** Diana Mutuku Mulatya, Joseph Olewe.

**Validation:** Diana Mutuku Mulatya, Vincent Were, Joseph Olewe.

**Visualization:** Diana Mutuku Mulatya, Vincent Were.

**Writing – original draft:** Diana Mutuku Mulatya, Joseph Olewe.

**Writing – review & editing:** Diana Mutuku Mulatya, Vincent Were, Japheth Mbuvi.

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
