## [Decision Letter · Decision Letter 0]

9 Nov 2020

PONE-D-20-24154

Willingness to pay for improvements in rural sanitation: Evidence from a cross-sectional survey of three rural counties in Kenya

PLOS ONE

Dear Dr. Mulatya,

Thank you for submitting your manuscript to PLOS ONE. After careful consideration, we feel that it has merit but does not fully meet PLOS ONE’s publication criteria as it currently stands. Therefore, we invite you to submit a revised version of the manuscript that addresses the points raised during the review process.

We look forward to receiving your revised manuscript.

Kind regards,

Raffaella Calabrese

Academic Editor

PLOS ONE

Journal Requirements:

2. Please ensure that you refer to Figure 6 in your text as, if accepted, production will need this reference to link the reader to the figure.

3. We note you have included tables to which you do not refer in the text of your manuscript. Please ensure that you refer to Tables 1 and 3 in your text; if accepted, production will need this reference to link the reader to the Tables.

4. Please upload a copy of Supporting Information S1 File which you refer to in your text on page 30.

"No authors have competing interest ".

We note that one or more of the authors are employed by a commercial company: Adaptive Management and Research Consultants(AMREC) Ltd.

Reviewers' comments:

Reviewer's Responses to Questions

**Comments to the Author**

1. Is the manuscript technically sound, and do the data support the conclusions?

Reviewer #1: Yes

Reviewer #2: Partly

2. Has the statistical analysis been performed appropriately and rigorously? 

Reviewer #1: Yes

Reviewer #2: I Don't Know

3. Have the authors made all data underlying the findings in their manuscript fully available?

Reviewer #1: Yes

Reviewer #2: Yes

4. Is the manuscript presented in an intelligible fashion and written in standard English?

Reviewer #1: Yes

Reviewer #2: Yes

5. Review Comments to the Author

Reviewer #1: Methods

1. Please provide a reference to the statement “…. with a glaring 50.8% of the population using unimproved sanitation and another 13.9% practicing open defecation rates”

2. The authors state that the study was a quantitative “economic evaluation … ” This is misleading to readers. A Contingent Valuation Method is not a form of an economic evaluation but a stated preference elicitation approach. I suggest deleting the economic evaluation part.

Results

1. Rephrase the statement “Of the 54 households with diarrhea (9.78%), most of them 6.51% were children aged >5 years.” I suppose this should be the households with a person with diarrea?

2. The figure numbers in the text need to be aligned. For instance the authors refer to figure 2 to be showing approximately 3% of the households being willing to pay for the market price for SAFI latrines whereas Figure two indicates the Map of study counties.

Reviewer #2: 1. The conclusion focused more on the recommendation. Author should first provide summary of result from data with supporting information before making recommendations

2. - Though the reviewer has expertise in the topic but is not specialized in using quantitative methodology (for example, the economic modeling used). Reviewer recommends having another reviewer with specialization in economic modelling

6. PLOS authors have the option to publish the peer review history of their article (what does this mean?). If published, this will include your full peer review and any attached files.

Reviewer #1: No

Reviewer #2: No

---

## [Author Response · Author response to Decision Letter 0]

2 Feb 2021

Journal requirements.

Below is the detailed response to all revisions requested for the manuscript.

Below is our response to the reviewer’s comments;

Reviewer #1: Methods

1. Please provide a reference to the statement “…. with a glaring 50.8% of the population using unimproved sanitation and another 13.9% practicing open defecation rates”

This is now provided as reference # 6 on the manuscript. Sourced from Kenya national bureau of statistics, Kenya Household Budgetary Survey 2015/2016

2. The authors state that the study was a quantitative “economic evaluation … ” This is misleading to readers. A Contingent Valuation Method is not a form of an economic evaluation but a stated preference elicitation approach. I suggest deleting the economic evaluation part.

Thanks, this has been deleted (Page 6)

Results

1. Rephrase the statement “Of the 54 households with diarrhea (9.78%), most of them 6.51% were children aged >5 years.” I suppose this should be the households with a person with diarrea?

We agree, this is corrected (Page 11) 

2. The figure numbers in the text need to be aligned. For instance the authors refer to figure 2 to be showing approximately 3% of the households being willing to pay for the market price for SAFI latrines whereas Figure two indicates the Map of study counties.

This have been corrected and figures aligned to the text updated

Reviewer #2: 1. The conclusion focused more on the recommendation. Author should first provide summary of result from data with supporting information before making recommendations

The conclusion has been reviewed accordingly and a summary of results from this study provided against each recommendation.

3. Though the reviewer has expertise in the topic but is not specialized in using quantitative methodology (for example, the economic modeling used). Reviewer recommends having another reviewer with specialization in economic modelling)

This comment was addressed by the editor

Other comments

4. The dataset should be anonymized and made fully available without restrictions

The attached data has been anonymized and will be made publicly available.

5. Under economic status, an author should provide details on wealth quantile categories. Expound on the principal content analysis and how each cluster was developed.

Household socioeconomic status (SES) indices were generated using multiple correspondence analyses using the following variables; occupation of household head, the primary source of drinking water, type of cooking fuel, ownership of household assets, and ownership of livestock. The households were categorized into five socioeconomic quintiles (wealth quintile) 

6. It's not clear why the author examined the WTP for SATO above the current market price of $10. Author to explain the relevance of this information.

The author examined willingness to pay for SATO pans and stools well above the market prices of $6 and $10 dollar to provide for an upper price range above the recommended retail price, within which customers would still be willing to pay for SATO products. This allows the author to simulate for a real market scenario, where product prices rise above recommended retail price. This mostly happens when middle men and brokers gain entry into the product supply chain, keep on making commercial gains.

---

## [Decision Letter · Decision Letter 1]

23 Feb 2021

Willingness to pay for improvements in rural sanitation: Evidence from a cross-sectional survey of three rural counties in Kenya

PONE-D-20-24154R1

Dear Dr. Mulatya,

We’re pleased to inform you that your manuscript has been judged scientifically suitable for publication and will be formally accepted for publication once it meets all outstanding technical requirements.

Kind regards,

Raffaella Calabrese

Academic Editor

PLOS ONE

Additional Editor Comments (optional):

Reviewers' comments:

Reviewer's Responses to Questions

**Comments to the Author**

1. If the authors have adequately addressed your comments raised in a previous round of review and you feel that this manuscript is now acceptable for publication, you may indicate that here to bypass the “Comments to the Author” section, enter your conflict of interest statement in the “Confidential to Editor” section, and submit your "Accept" recommendation.

Reviewer #1: All comments have been addressed

2. Is the manuscript technically sound, and do the data support the conclusions?

Reviewer #1: Yes

3. Has the statistical analysis been performed appropriately and rigorously? 

Reviewer #1: Yes

4. Have the authors made all data underlying the findings in their manuscript fully available?

Reviewer #1: Yes

5. Is the manuscript presented in an intelligible fashion and written in standard English?

Reviewer #1: Yes

6. Review Comments to the Author

Reviewer #1: (No Response)

7. PLOS authors have the option to publish the peer review history of their article (what does this mean?). If published, this will include your full peer review and any attached files.

Reviewer #1: No

---

## [Editor Report · Acceptance letter]

12 Apr 2021

PONE-D-20-24154R1 

Willingness to pay for improvements in rural sanitation: evidence from a cross-sectional survey of three rural counties in Kenya 

Dear Dr. Mulatya:

I'm pleased to inform you that your manuscript has been deemed suitable for publication in PLOS ONE. Congratulations! Your manuscript is now with our production department. 

Kind regards, 

on behalf of

Dr. Raffaella Calabrese 

Academic Editor

PLOS ONE